# Mortality Risk Stratification in Emergency Surgery for Obstructive Colon Cancer—External Validation of International Scores, American College of Surgeons National Surgical Quality Improvement Program Surgical Risk Calculator (SRC), and the Dedicated Score of French Surgical Association (AFC/OCC Score)

**DOI:** 10.3390/ijerph192013513

**Published:** 2022-10-19

**Authors:** Raul Mihailov, Dorel Firescu, Georgiana Bianca Constantin, Oana Mariana Mihailov, Petre Hoara, Rodica Birla, Traian Patrascu, Eugenia Panaitescu

**Affiliations:** 1Clinic Surgery Department, Dunarea de Jos University, 800216 Galati, Romania; 2Morphological and Functional Sciences Department, Dunarea de Jos University, 800216 Galati, Romania; 3Clinic Medical Department, Dunarea de Jos University, 800216 Galati, Romania; 4General Surgery Department, Carol Davila University, 050474 Bucharest, Romania; 5Medical Informatics and Biostatistics Department, Carol Davila University, 050474 Bucharest, Romania

**Keywords:** external validation, prediction, emergency care, obstructive colon cancer

## Abstract

Background: The increased rates of postoperative mortality after emergency surgery for obstructive colon cancer (OCC) require the use of risk-stratification scores. The study purpose is to external validate the surgical risk calculator (SRC) and the AFC/OCC score and to create a score for risk stratification. Patients and methods: Overall, 435 patients with emergency surgery for OCC were included in this retrospective study. We used statistical methods suitable for the aimed purpose. Results: Postoperative mortality was 11.72%. SRC performance: strong discrimination (AUC = 0.864) and excellent calibration (11.80% predicted versus 11.72% observed); AFC/OCC score performance: adequate discrimination (AUC = 0.787) and underestimated mortality (6.93% predicted versus 11.72% observed). We identified nine predictors of postoperative mortality: age > 70 years, CHF, ECOG > 2, sepsis, obesity or cachexia, creatinine (aN) or platelets (aN), and proximal tumors (AUC = 0.947). Based on the score, we obtained four risk groups of mortality rate: low risk (0.7%)—0–2 factors, medium risk (12.5%)—3 factors, high risk (40.0%)—4 factors, very high risk (84.4%)—5–6 factors. Conclusions: The two scores were externally validated. The easy identification of predictors and its performance recommend the mortality score of the Clinic County Emergency Hospital of Galați/OCC for clinical use.

## 1. Introduction

Colorectal cancer (CRC) is the third most common cancer worldwide. It is the third most common cancer in men and the second in women. There were over 1.9 million new cases of colorectal cancer in 2020, with Hungary having the highest overall rate, followed by Slovakia. Romania was not in the top 10 countries in terms of incidence but was ranked ninth among countries with a high global colorectal cancer mortality rate in 2020 [1].

Acute symptoms, which lead to an emergency surgery, occur in up to 30% of patients with CRC [2], and 8–13% of patients develop acute bowel obstruction, one of the major complications of this disease [3]. Dealing the patients with obstructive colon cancer (OCC) involves a rapid approach to solve two problems: first, bowel obstruction, which leads to ischemia, necrosis, and perforation [4,5], and second, treating cancer with curative intent whenever possible.

Emergency surgery for OCC is associated with worst postoperative results compared with elective interventions, with an 8–15% mortality rate and even higher (30%) in elderly patients [6,7,8].

In these patients, with a high postoperative mortality rate, preoperative assessment of surgical risk based on risk scores is required by integrating and interpreting a large amount of clinical information. Many surgical risk scores have been developed to estimate postoperative mortality and, less commonly, specific complications. The most common score used by anesthesiologists—the classification of physical status developed by the American Society of Anesthesiologists (ASA score)—is based on the subjective assessment of the patient’s preoperative health and shows a good correlation with the risk of postoperative mortality [9].

In 2013, the American College of Surgeons developed a surgical risk calculator (SRC) on a sample of NSQIP (National Surgical Quality Improvement Program) in an attempt to accurately stratify patients based on predicting individual risks of mortality and postoperative complications [10]. For the prediction of postoperative mortality at 30 days, it proved to be an excellent tool for many groups of patients proposed for performance validation [11]. However, few studies have validated the calculator in an external cohort of patients undergoing emergency surgery for OCC, and some studies have questioned its value in external populations [12,13]. In 2018, a French study validated for the first time on 69 patients the performance of the SRC in patients with OCC requiring emergency surgery, supporting the possibility of using this score worldwide [14].

Several authors have reported the detection of risk factors or logistic models for the prediction of postoperative mortality after colorectal surgery (French Surgical Association (AFC) score, Association of Coloproctology of Great Britain and Ireland score, and CCR-CARESS score, colorectal POSSUM) [15,16,17,18]. Most of the studies observed that emergency surgery for CRC was an independent factor for postoperative mortality [16,19,20]. However, the studies that disseminated these scores used heterogeneous cohorts in terms of diagnosis, including both patients with CRC as well as non-malignant diseases. To optimize surgical management, it would be advisable to have the most precise tools available for a specific pathology and clinical situation. In 2019, based on a cohort study, a dedicated AFC/OCC score was proposed for these patients but has not yet been followed by an external validation [21].

The purpose of the study was to evaluate the performance of the SRC and AFC/OCC scores on our population (external group of patients) by determining the prediction accuracy of the mortality risk as well as the creation of a logistic regression model and an own score that allows the stratification of the risk of postoperative mortality in these patients.

## 2. Materials and Methods

### 2.1. Data Collection

We included in the study group the patients with OCC who presented in the emergency room during a period of 10 years, from 2008 to 2017, at the “Sf. Apostol Andrei” Emergency County Clinical Hospital in Galati and who were operated in emergency in the 1st and 2nd Clinic of General Surgery.

The observation sheets, the analysis reports, and the operative protocols were retrospectively investigated. The clinical and epidemiological data monitored were age, gender, personal antecedents, comorbidities, functional status expressed using the classification of Eastern Cooperative Oncology Group (ECOG), presence of cachexia or obesity, smoking status, septic condition, and the ASA class assessed by the anesthesiologist. The biological and imaging data analyzed were the values of leukocytes (WBC), platelets, hemoglobin, glycemia, creatinine, proteins, albumins, the presence of electrolyte disorders, acidosis, coagulation disorders, as well as the results of abdominal radiographs and computed tomography.

Inclusion criteria: adult patients with OCC, admitted without previous investigations or diagnosis, with emergency open surgery, treated in that period, and with complete recorded data. Exclusion criteria: patients who refused surgical intervention, those with scheduled operations, those who underwent diagnostic laparotomy with or without biopsy, those with incomplete recorded data, and those in whom malignancy was not confirmed.

### 2.2. Study Design

We performed an external validation study of two scores of postoperative mortality: SRC and AFC/OCC score, on a group of patients with emergency surgery for OCC, which analyzed the reliability of these scores. Secondly, we performed logistic regression analysis to identify predictive factors, based on which we created a model and developed a performing score for stratifying the risk of postoperative mortality.

### 2.3. Variables and Definitions

The age was entered in the online risk calculator according to the age range from the SRC (https://riskcalculator.facs.org/RiskCalculator/). The dichotomization with the threshold of ≥75 years was performed for the calculation of the AFC/OCC score. Based on the calculated threshold value (>70 years) of the observed mortality, the variable was dichotomized.

The comorbidities were evaluated as qualitative variables but also in the form of scores: Davies, Charlson, and Charlson adjusted with age, for which threshold values were determined (>2, >2, >9), based on which the variables were dichotomized.

Cardiovascular comorbidities included the history of ischemic heart disease, congestive heart failure (CHF), cardiac or vascular surgery, valvular disease, rhythm disorders, hypertension (HTN), or peripheral artery disease. The presence of the following pathology in the patients: HTN, CHF, and hemodynamic failure—defined as all patients who got vasopressors preoperatively or as the appearance of one of the following: systolic arterial pressure < 90 mmHg, mean arterial pressure < 70 mmHg, heart rate (HR) > 120/min, urine output < 15 mL/h, and/or confusion—were registered as qualitative variables.

Pulmonary comorbidities were noted in the patients with asthma, chronic obstructive pulmonary disease, emphysema, and a history of pulmonary tuberculosis or lung carcinoma. The presence of dyspnea and a history of severe chronic obstructive pulmonary disease (COPD) were recorded.

Renal comorbidities were registered in patients with a history of chronic kidney failure, nephrectomy or polycystic kidney, and kidney transplantation. Acute renal failure was noted for the calculation of the SRC prediction.

Neurological comorbidities were recorded in patients with a cerebrovascular event, with or without neurological deficit, history of transient ischemic attack, dementia, neurodegenerative disease, or psychiatric disorder.

Patients with diabetes mellitus type I and II disease and insulin-dependent forms were recorded for use in SRC. History of cirrhosis or viral hepatitis was included as hepatic comorbidities.

Smoker status was recorded and used to calculate SRC prediction. For the assessment of obesity and cachexia, we used the criteria recommended by the World Health Organization based on the body mass index (BMI), obtained by dividing the body mass to the squared of the body height. Cachexia and obesity were recorded in case of a BMI < 18.5 kg/m^2^, respectively >30 kg/m^2^.

The ECOG (Eastern Cooperative Oncology Group) performance status was assessed from 0 to 4. To calculate the SRC prediction, we used grade 0 for functional status—independent, grades 1, 2, 3—partially dependent, and grade 4—totally dependent. We calculated the threshold value for postoperative mortality, and we dichotomized this variable (>2).

ASA score was correlated with the questions from SRC: 2—mild systemic disease, 3—severe systemic disease, 4—severe systemic disease/constant threat to life, and 5—moribund/not expected to survive the surgery. For AFC/OCC score calculation, the qualitative variable was dichotomized according to the score factor (≥3), which was equal to the obtained cutoff value (>2) concerning the observed mortality.

The laboratory values (white blood cells (WBC), anemia, platelets number, blood sugar, creatinine, electrolyte disturbances, acidosis, and coagulation disorders) were evaluated as qualitative variables, values outside the normal range being considered abnormal values (aN).

Septic status was recorded in patients with OCC with 2 or more of the following criteria: temperature > 38 °C or <36 °C; HR > 90/min; respiratory rate (RR) > 20/min or partial pressure of CO2 (PaCO2) < 32 mmHg; and WBC > 12,000 cells/mm^3^; it was necessary to calculate the SRC prediction.

The intraoperative data included the location of the tumor, the presence of local invasion or metastases, the presence of ischemic lesions of the distended colon, and the presence of peritonitis. The occurrence of intraoperative complications was assessed as qualitative variable.

Tumor location was dichotomized into proximal location up to the level of the splenic flexure or distal—distal to the splenic flexure—according to the categorical variable of the AFC/OCC score. We dichotomized the variable according to the anatomical segments of the colon into 8 groups: cecum (C), ascending colon (A), hepatic flexure of the colon (HF), transverse colon (T), splenic flexure of the colon (SF), descending colon (D), sigmoid (S), and rectosigmoid junction (RSJ).

The type of surgical intervention was assessed according to the current code of procedural terminology (CPT) used by the SRC: 44,320—diverting stoma, 44,143—Hartman resection, 44,140—partial colectomy with anastomosis, 44,144—partial colectomy with protective ileostomy, 44,150—total colectomy with ileoproctostomy, 45,399—intestinal bypass—unlisted procedure, colon. For the AFC/OCC score, this variable was dichotomized in patients with a stoma (diverting stoma, Hartman resection, partial colectomy with protective ileostomy) and patients without a stoma—partial or total colectomy with anastomosis or colic bypass. For our analysis, we dichotomized patients with a colostomy (DC), Hartman resection (HP), colic bypass (BP), partial resection with anastomosis (PCA), partial resection with protective ileostomy (PCAI), and total resection with anastomosis (TCA).

The duration of the surgical intervention was dichotomized into ≥150 min and <150 min according to the requirement of the AFC/OCC score. Initially, for this variable, we recorded 10 groups of values from 60 min to 300 min. We calculated the threshold value of the duration of the intervention concerning the postoperative mortality and dichotomized the variable according to this (>120 min).

### 2.4. Outcome Parameters

The main outcome parameters were: performance evaluation in terms of discrimination and calibration of SRC prediction and AFC/OCC score.

Secondary outcome: the creation of our score, evaluation of its performance, and stratification of the mortality risk in the patients included in the study.

### 2.5. Predicted Data: SRC

Data were collected for the 20 predictors of SRC: patient demographics (age, sex (gender), functional and smoking status, weight, height) and comorbidities (ASA class, HTN, diabetes, CHF, COPD, dyspnea, ventilator dependence, acute renal failure, dialysis, ascites, steroid use, systemic sepsis, and disseminated cancer). Surgical procedures were reviewed and matched to CTP used by the computer. The question “Are there other potentially appropriate treatment options?” was set to “none”, and the “Surgeon risk adjustment” was set to “No adjustment required” for all cases. Data were entered into the SRC online calculator (http://riskcalculator.facs.org/RiskCalculator/, accessed on 1 August 2022) and predicted mortality risks were recorded for each patient.

### 2.6. Predicted Data: AFC/OCC Score

We calculated the score for each patient by summing up the risk factors present: age ≥ 75 years, ASA score ≥ 3, pulmonary comorbidities, proximal colon tumors, and hemodynamic failure, and we recorded the mortality risk predicted by the score according to the score’s recommendations: 0–1.3%, 1–4.2%, 2–7.0%, 3–16.9%, 4–27.5%, and 5–50.0% [21].

### 2.7. Statistical Methods

#### 2.7.1. External Validation Analyses for SRC and AFC/OCC Score

We appreciated the performance of the SRC and AFC/OCC score using the receiver operating characteristics (ROC) curves, the observed mortality/estimated mortality ratio (O/E ratio), the Hosmer–Lemeshow goodness-of-fit test (H-L test), the calibration curves, and the Brier score (BS). We generated an ROC curve, and we calculated the area under the curve (AUC) for assessing the prediction accuracy concerning discrimination, whereby the area of 0.5 indicated discriminating power not better than chance, 0.6–0.69 discrimination was considered poor, adequate with an AUC of 0.7–0.79, strong with an AUC of 0.8–0.89, excellent with an AUC of 0.9–1.0, and the area of 1.0 indicated perfect discriminating power. A score is considered reliable within a clinical setting if its AUC value is ≥0.75 [22].

We determined the mean predicted postoperative mortality and correlated it to the observed mortality. The observed/expected ratio of 1 represents perfect accuracy of calibration, a ratio < 1 indicates over prediction, and a ratio >1 indicates under prediction. Calibration was further evaluated using the H–L test, defining a lack of fit as a *p*-value. In addition, for calibration, we graphically represented the predicted score probabilities versus the observed ones. The closer the represented curve is to the straight line—corresponding to the identity function (the estimated probabilities are always the same as the observed ones)—the better the calibration.

BS was used because of its ability to reflect both discrimination and calibration simultaneously. We determined BS as the average squared difference between the predicted probabilities of events and the observed events (0 for nonevents; 1 for events). A BS of 0 indicates perfect prediction, whereas a BS of 1 indicates the poorest prediction; a score ≥ 0.25 is considered non-informative [23]. We correlated BS to the null model Brier, which was calculated by assigning to every patient the overall observed rate of each of the outcomes as the probability of experiencing an event. The smaller the BS is, comparing with the null model BS, the higher the accuracy of the score.

Further, for the AFC/OCC score, we utilized specific statistics for measuring the performance of a score based on external validation [14]. We calculated a simple separation parameter (PSEP), which is the difference PSEP = (p worst − p best), where p worst is the mortality rate of patients in the group with the worst prognosis, and p best is the mortality rate of patients in the group with the best prognosis. The analogy between the PSEP value calculated in the training sample and the value calculated in the validation sample permitted us to appreciate whether the discriminatory ability of the score has been evaluated too optimistically or not [24].

#### 2.7.2. Prediction Factors and Performance of Own Score

The categorical variables were synthesized as frequencies and percentages, and the quantitative variables were as medians and quartiles. We performed statistical correlations, indicating the *p*-value with the Pearson chi-square, likelihood ratio, and Fisher’s exact test for categorical variables and the Mann–Whitney test for quantitative variables. For continuous variables, we used ROC curves to identify a threshold value. To identify the predictive factors, a univariate logistic regression analysis was performed, specifying the estimated relative risk (OR) and its confidence interval (95.0%CI). Multivariate logistic regression analysis was performed to determine a prediction model for postoperative mortality, specifying for each predictor OR and 95.0%CI, and we evaluated the performance of the model. For the internal validation of the model, we used the bootstrapping method. We calculated the values of the Cox and Snell and Nagelkerke coefficients, the H-L test (*p* > 0.05 showed good agreement), c-statistic, and BS.

A new clinical score was built based on the predictors revealed by the model. The performance of this score was analyzed by generating an ROC curve with the calculation of the AUC. We also aimed to assess the optimal threshold value for mortality risk prediction (low risk, high risk), the sensitivity, and the specificity. The stratification of mortality risk depending on the number of factors present in one patient (low risk, medium risk, high risk, and very high risk) was performed using logistic regression analysis.

Statistical conclusions were drawn using *p* < 0.05 as a significant difference threshold for all calculations performed. SPSS version 23.0, R version 4.1.1, and Med Calc version 20.111 were used in the statistical analysis.

## 3. Results

The study group included 435 patients with an average age of 68.44 ± 11.748, ranging from 27 to 92 years.

Table 1 summarizes the demographic data, comorbidities, and laboratory values of all patients; statistical correlation with postoperative mortality; and univariate logistic regression analysis. In the group, 38.85% were women, 66.2% of the patients were classified with ASA class 3 or 4, and 9.19% presented preoperative sepsis criteria. The most frequent comorbidities were HTN (25.51%), kidney diseases (8.27%), and diabetes (10.57%). Functional dependence was observed in 81.83% of patients and cachexia in 22.06%. Among the laboratory values, anemia was present in 64.36% and pathological values of WBC in 45.28%. The tumor location was proximal in 70.34% of the patients, and 21.37% had metastases. In 57.47% of the patients, procedures associated with a stoma were performed (DC, HP, PCAI) and in the rest BP, PCA, or TCA. Postoperative morbidity was 22.29% (97/435). Postoperative mortality at 30 days was 11.72% (51/435).

The causes of death were as follows: septic shock (22) (16 were already admitted with sepsis), CHF (16), heart attack (2), anastomotic leakage (6) (in five patients after PCA and one after BP), acute pulmonary failure (3), acute renal failure (1), and intestinal obstruction (1).

Postoperative mortality related to the surgical procedure was as follows: 13/140 patients with PCA, with 5 of them associated with anastomotic leakage (9.3%); 6/42 patients with BP, with 1 through the anastomotic fistula (14.3%); 21/169 patients with HP (12.4%); 9/74 patients with DC (12.2%); 2/7 patients with PCAI (28.6%); and 0/3 patients with TCAI.

### 3.1. SRC Score Reliability

The average mortality rate estimated by SRC was 11.80%. The univariate analysis confirmed that mortality predicted by the SRC was significantly associated with the observed mortality (OR = 1.10, 95%CI = 1.07–1.13, *p* = 0.000000).

The ROC curve generated to evaluate the score’s discrimination power revealed an AUC = 0.864, (*p* < 0.001), which indicates a strong discrimination potential. With the optimal cutoff value > 12%, SRC identified 88.2% of deceased patients and 71.1% of survivors. Patients with an estimated risk > 12% constitute the group with a high risk for postoperative mortality (Figure 1).

The SRC indicated an excellent calibration of prediction for mortality (11.80% predicted versus 11.72% observed), with the O/E ratio very close to 1 (0.993). When analyzing SRC performance in predicting postoperative mortality depending on the surgical procedure, SRC underestimated mortality in patients with PCA (6.06% versus 9.30%), PCAI (24.49% versus 28.6%), and BP (13.71% versus 14.3%) and overestimated mortality in patients with HP (13.43% versus 12.4%) and DC (17.16% versus 12.2%). The H-L test indicated a good calibration with *p* = 0.6422 (>0.05). The calibration graph is represented in Figure 2; the graph confirms the fit between the predicted and observed data, as the points of the graph are close to the straight line. BS was 0.080 < null model (0.103) and less than 0.25, indicating a good accuracy of the prediction.

### 3.2. AFC/OCC Score Reliability

The average mortality value estimated by the AFC/OCC score was 6.93%. The univariate analysis confirmed that mortality predicted by the AFC/OCC score was significantly associated with the observed mortality (OR = 1.17, 95%CI = 1.11–1.22, *p* = 0.000000).

The ROC curve generated to evaluate the performance of the score revealed an AUC = 0.787, (*p* < 0.001), which indicates an adequate power of discrimination. With the optimal cutoff value > 4.2%, the score identified 86.3% of the cases that died and only 59.4% of the survivors. Patients with a predicted risk > 4.2% constitute the group with a high risk for postoperative mortality (Figure 3).

The AFC/OCC score underestimated the mortality (6.93% predicted versus 11.72% observed) with O/E > 1. The performance of the score in predicting postoperative mortality depending on the surgical procedure; the score underestimated mortality in all types of procedures: PCA (6.93% versus 9.30%), PCAI (11.3% versus 28.5%), BP (8.25% versus 14.3%), HP (6.28% versus 12.4%), and DC (7.54% versus 12.2%).

The mortality rates obtained in the validation group were as follows: in patients without risk factors, 3.17% (2/63) compared to 1.3% (4/313) in the training sample [21]; with one factor, 2.32% (4/172) compared to 4.2% (20/472); with two factors, 15.74% (20/127) compared to 7% (26/372); with three factors, 28.81% (17/59) compared to 16.9% (34/201); and with four factors, 57.14% (8/14) compared to 27.5% (14/51). We did not identify in the group any patients with five factors, and in this risk group from the training sample, mortality was 50% (3/6) [21].

We calculated the PSEP on the validation sample (54.82%) and compared it with the value obtained on the training sample (48.7%) [21].

The H-L test did not indicate a good calibration of the prediction of the score (*p* = 0.0005054) (<0.05). The calibration graph is represented in Figure 4; the graph confirms the mismatch between the predicted and observed data, as the points of the graph are not close to the straight line. BS was 0.095 < null model (0.103) and less than 0.25, which confirms the accuracy of the prediction.

### 3.3. Postoperative Mortality Predictors

The 26 risk factors of postoperative mortality were: age > 70 (OR = 4.75, 95%CI = 2.31–9.76, *p* = 0.000022), CHF (OR = 4.13, 95%CI = 1.47–11.6, *p* = 0.006799), hemodynamic failure (OR = 6.79, 95%CI = 3.64–12.7, *p* = 0.000000), pulmonary comorbidities (OR = 5.76, 95%CI = 2.23–14.9, *p* = 0.000000), COPD (OR = 7.79, 95%CI = 1.07–56.6, *p* = 0.042320), renal comorbidities (OR = 3.94, 95%CI = 1.80–8.62, *p* = 0.000565), acute renal failure (OR = 15.6, 95%CI = 1.39–176, *p* = 0.025893), neurological comorbidities (OR = 4.92, 95%CI = 2.05–11.8, *p* = 0.000363), obesity (OR = 2.83, 95%CI = 1.24–6.42, *p* = 0.012729), cachexia (OR = 6.88, 95%CI = 3.71–12.8, *p* = 0.000000), ECOG > 2 (OR = 5.42, 95%CI = 2.95–9.98, *p* = 0.000000), Davies score > 2 (OR = 3.28, 95%CI = 1.76–6.13, *p* = 0.000184), Charlson score > 2 (OR = 7.20 95%CI = 2.54–20.4, *p* = 0.000201), age-adjusted Charlson score > 9 (OR = 7.12, 95%CI = 2.96–17.1, *p* = 0.000011), sepsis (OR = 8.97, 95%CI = 4.37–18.4, *p* = 0.000000), WBC (abnormal values) (OR = 9.49, 95%CI = 4.16–21.6, *p* = 0.000000), anemia (OR = 3.94, 95%CI = 1.73–9, *p* = 0.001098), platelets (abnormal values) (OR = 9.75, 95%CI = 5.11–18.6, *p* = 0.000000), glycemia (abnormal values) (OR = 3.07, 95%CI = 1.67–5.64, *p* = 0.000296), creatinine (abnormal values) (OR = 4.71, 95%CI = 2.46–9.01, *p* = 0.000003), electrolyte disturbances (OR = 4.28, 95%CI = 2.33–7.85, *p* = 0.000003), acidosis (OR = 5.13, 95%CI = 2.79–9.43, *p* = 0.000000), coagulation disorders (OR = 5.27, 95%CI = 2.70–10.3, *p* = 0.000001), ASA > 2 (OR = 6.97, 95%CI = 2.46–19.8, *p* = 0.000258), operative time > 120 min (OR = 3.09, 95%CI = 1.70–5.62, *p* = 0.000209), and peritonitis (OR = 3.10, 95%CI = 1.05–9.09, *p* = 0.039192) (Table 2).

Nine independent predictors of postoperative mortality at 30 days were identified within a multivariate logistic analysis model: age > 70 years (OR = 12.9, 95%CI = 3.62–46, *p* = 0.000), CHF (OR = 13.5, 95%CI = 3.45–52.5, *p* = 0.000), ECOG > 2 (OR = 6.69, 95%CI = 2.65–16.9, *p* = 0.000), sepsis (OR = 14.0, 95%CI = 4.21–46.9, *p* = 0.000), obesity (OR = 6.80, 95%CI = 1.90–24.3, *p* = 0.003), cachexia (OR = 5.19, 95%CI = 2.08–12.9, *p* = 0.000), abnormal value of platelets (OR = 11.6, 95%CI = 4.30–31.3, *p* = 0.000), abnormal value of creatinine (OR = 3.39, 95%CI = 1.41–8.13, *p* = 0.006), and proximal tumors (OR = 2.95, 95%CI = 1.17–7.42, *p* = 0.021) (Table 2).

The calculated AUC for this model was 0.947 (95%CI = 0.922–0.972, *p* < 0.001), which showed an excellent discriminatory power of the predicted values of postoperative mortality. With the optimal cutoff value > 14%, the model identified 88.2% of the cases that died and 87% of the survivors. Patients with a mortality risk > 14% constitute the high-risk group (Figure 5).

The model was well calibrated with no significant difference between the predicted and observed probabilities (H-L test, *p* = 0.970). This model explained 31.10% of the observed data (Cox and Snell R^2^) and up to 60.50% (Nagelkerke R^2^). We determined a c-statistic that was 0.947 and equal to AUC.

For internal validation, we used the bootstrapping method (Figure 6).

The *Y*-axis represents the actual probability of postoperative death. The *X*-axis represents the predicted probability of postoperative death. The ideal line is a perfect prediction model. The apparent line represents the performance of the model, and a close match to the ideal line is a good prediction, as seen in our calibration graph. The BS value is 0.053 < null model (0.103) and less than 0.25, which confirms the accuracy of the prediction.

A clinical score (called Mortality Score—Clinic County Emergency Hospital of Galați/OCC) was built based on the nine predictors, in which we assigned 1 point to each of the predictors. Thus, the score for the patients in the group varied between 0 and 6 points (the maximum number of factors present in one patient was 6). To evaluate the performance of the score, we generated an ROC curve. The calculated AUC for this model was 0.932 (95%CI = 0.922–0.972, *p* < 0.001), which showed an excellent discriminatory power of the values predicted for postoperative mortality. With the optimal cutoff value > 2 factors, the score identified 96.1% of the cases that died and can identify 73.7% among the survivors. Patients with more than two factors for mortality risk constitute the high-risk group (Figure 7).

Table 3 shows the variations in mortality rates and OR values according to the number of predictive factors identified in the model. Patients with a maximum of one risk factor had a mortality rate of 0% (0/172), those with two factors 0.7% (2/113), low risk; those with three factors 12.5% (10/80), medium risk; those with four factors 40% (18/45), high risk; those with five risk factors 81.8% (18/22); and 100% (3/3) for those with six risk factors, very high risk.

## 4. Discussion

The most important thing when obtaining consent for emergency surgery is informing the patient of the underlying pathology, available treatments, and risks and benefits of each option. The surgeon presents the imperative nature of the intervention, which can save a patient’s life, but also the postoperative complications that can often occur and are extremely significant, leading to postoperative death. This risk along with others—postoperative complications, prolonged hospital stay, or re-intervention—often weighs enormously in the patient’s decision to accept the surgical intervention. Traditionally, surgeons have estimated this risk based on their clinical judgment and personal experience. The introduction of risk calculators or surgical risk scores aims to provide patients with a more realistic and personalized prediction of the risks associated with surgery, based on which to decide, along with the surgeon, on the optimal surgical procedure and appropriate postoperative care. Surgical risk stratification with the help of these tools is very important because postoperative deaths occur mainly in the high-risk patient group. This measure of postoperative risk stratification is all the more important the higher the postoperative mortality rate after a certain type of intervention. This category also includes emergency surgery in general [25,26] and patients with OCC in particular [6].

The postoperative mortality in the study group was 11.72%, which is lower compared to the results published by other studies but in smaller groups of patients: 18.8% in a group of 109 patients with OCC [27] or 29% in a group of 75 patients with emergency colorectal operations [28].

Among the multitude of scores proposed in the literature, for the prediction of postoperative mortality in patients with OCC who were undergoing emergency surgery, we chose to evaluate the performance of two scores, using the group of patients as an external validation group, a universal risk calculator developed based on the data of a large cohort of patients (SRC), and a dedicated score for patients with OCC (AFC/OCC score), developed on 1983 patients. SRC is easy to use and compatible with the type of surgical procedures of the patients in the group (all patients in the group had open operations), with rather divergent results from the literature on external validation for emergency surgery or emergency colorectal interventions [13,29,30,31,32]; the study represented the first evaluation of this score on a cohort of patients from Romania and the second external validation study on OCC patients undergoing emergency surgery. The AFC/OCC score is easy to use, and all risk factors can be identified preoperatively, which is a great advantage for establishing therapeutic options but has no external validation study to date.

In the SRC validation of the study, we obtained a strong predictive accuracy in terms of discrimination ability, expressed by an AUC of 0.864, which are similar results to a study on a group of 301 patients with urgent colectomy for various conditions: AUC of 0.8451 [33] or AUC of 0.887 as reported by an Italian study on a group of 317 patients with emergency operations that also included 86 patients with intestinal obstruction [34]. This performance is lower than those obtained in the training sample that used a group of 1,414,006 patients (c-statistic = 0.944) or for the 41,784 patients with colon neoplasm (c-statistic = 0.8566) [10]. The first external validation study of SRC on 69 patients with OCC does not report the AUC value [14].

Regarding the calibration of the SRC prediction, we obtained excellent predictability for postoperative mortality (11.80% predicted versus 11.72% observed) with an O/E ratio of 1.006, unlike a study on patients with emergency operations that present an O/E ratio of 2.48 between observed deaths (11) and those estimated by the SRC (4.42) [35]; in another study on 75 patients with emergency colectomy, the SRC predicted a mortality rate of 26% versus 29% for the observed mortality rate [28] or another study on patients with OCC in which SRC overestimated the mortality rate: 9.8% predicted versus 8.7% observed [14].

Regarding the type of interventions performed, these reports were different: SRC underestimated mortality after PCA, PCAI, and BP and overestimated mortality after HP and DC, and these discrepancies are explained by the small number of deaths in each subgroup. Obtaining better results than those estimated in patients with CD or HP is explainable because these patients were in a more severe condition, and the choice of the surgical procedure took this aspect into account. The underestimation of mortality in patients with anastomoses can also be explained by the type of intervention that involves assuming complications associated with anastomoses that can sometimes lead to death in patients who presented in a better biological state. The data from the literature regarding the external validation studies of this score present similar situations where the analysis of the subgroups in the batch shows deviations of the prediction rate; the prediction is all the better the more the batch is similar to the NSQIP sample, which contains case mixes [13].

The BS obtained on the batch was 0.080, which is lower than the null model BS (0.103), which indicates the performance of the score. The value of BS obtained was close to the results published by an Italian study (0.072) [34] but higher than the one obtained for the prediction of mortality in the model training group (0.011), the subgroup of patients with colon cancer from this group (0.0184) [10], or the results of the first external validation study of SRC on 69 patients with OCC (0.058) [14]. The obtained results confirm the reliability of SRC in predicting postoperative mortality for patients with OCC operated in an emergency hospital in Romania.

In the validation study of the AFC/OCC score, we obtained an adequate predictive accuracy of the discrimination power expressed by the AUC of 0.787 > 0.75, being considered reliable within the clinical setting. In comparison with the training sample, where a model with an AUC of 0.80 was obtained, with a sensitivity of 69% and a specificity of 82%, in our batch, we obtained a sensitivity of 86.3% and a specificity of 59.4%—close to 60%—a value taken by some authors as the indispensable lower limit to consider the threshold value (4.2%) informative [34].

Regarding the calibration of the AFC/OCC score, it underestimated both the mortality rate for the entire batch (6.93% predicted versus 11.72% observed with an O/E ratio of 1.69) as well as the one reported depending on the surgical procedure used. The H-L test indicated *p* < 0.05, and the aspect calibration graph indicated low calibration, but BS (0.095) confirmed the performance of the model (0.095 < null model BS (0.103) and less than 0.25). Evaluating by BS is considered by some authors [10] to be more appropriate in assessing the performance prediction of a score. The H-L and BS tests were not calculated in the training sample of the score [21]. The PSEP value on the validation batch (54.82), higher than that obtained on the training batch (48.7), denotes that the discriminatory ability of the AFC/OCC score from the training sample is fully preserved in this external validation sample [21]. This method of evaluating the performance of a score in the external validation group was also used by other authors [15].

By the obtained results, using the patient group as an external validation sample of the SRC and the AFC/OCC score, we consider that we have fulfilled the first objective of the study. From the point of view of the surgeon, who is found in the situation of choosing which of the two prediction methods to use for estimating the risk of mortality, for a patient with OCC requiring emergency surgery, he would probably choose the AFC/OCC score even though the performance is lower, and it does not offer a personalized risk but only the inclusion in a risk group. It is the dedicated score for patients with OCC who require emergency surgery: the factors necessary to calculate it are identifiable preoperatively, and it is simple to apply. Mortality prediction by SRC is personalized and statistically more efficient, but unlike elective surgery, in which the intervention performed usually coincides with the proposed one, in emergency surgery, quite often the final surgical decision is taken intraoperatively. Therefore, in the estimation of mortality through this score, the type of intervention expected and entered in the computer preoperatively may differ from the one performed. More studies have reported multivariate analysis of predictive factors for postoperative mortality in patients operated on for OCC [15,27,36,37,38,39,40,41,42,43,44]. Some older studies have already been reported in Manceau’s study [21] published in 2019 [27,36,37,38]. We have identified in the literature other more recent studies that report these aspects [39,41,42,43,44].

Alves et al., in a prospective French multicenter study on a group of 1049 patients with colonic resections for cancer (75%) or diverticulitis, reported a mortality rate of 4.6% and the following predictors of postoperative mortality: age older than 70 years, neurological comorbidities, body weight loss > 10% in <6 months, and emergency surgery [15].

The European Society of Coloproctology reported in a study conducted on a group of over 2000 patients operated on with CRC as predictors for mortality age > 71 years, emergency surgery, open approach, ASA ≥ 3, mechanical anastomosis, intraoperative complications, use of statins, and abnormal creatinine levels [39].

Longo et al. identified sixteen predictive factors with a postoperative mortality rate of 5.7% in a group of patients with resections for CRC: ASA ≥ 3, ascites, sodium > 145 mEq/L, platelets < 150,000 mm^3^, potassium < 3.5 mEq/L, sodium < 135 mEq/L, use of steroids, neurological deficits after stroke, disseminated cancer, abnormal values of serum urea nitrogen, alkaline phosphatase, sensory impairment, and low values of serum albumin [40].

Teloken et al., in a study conducted on 249 patients with emergency colon resections, reported after logistic regression analysis that age > 65 years, ASA 4, and cancer were independent predictive factors of mortality at 30 days and a mortality rate of 6.8% [41].

Another study on a group of 776 patients with obstructive right colon cancer reported a postoperative mortality rate of 10% and the following predictive factors of postoperative mortality: age > 70 years, ASA > 3, and hemodynamic instability at admission [42].

In a group of 40 patients with emergency colorectal operations, the authors reported mortality of 12.5%, with the following risk factors: higher ASA score, absence of screening colonoscopy, and postoperative medical complications [43].

On a sample of NSQIP of 11,698 patients with colectomies, of which 29.1% were emergency interventions, postoperative mortality was significantly higher in emergency colectomy 872/3409 (25.6%) versus elective colectomy 567/8289 (6.8%), with risk factors being higher ASA classification and more comorbidities [44].

The mortality rate observed in our study (11.72%) is compared to the data presented closer to the lower limit of the range (6.9–25.6%).

In the study group, we identified 26 risk factors for postoperative mortality: age > 70 years, CHF, hemodynamic failure, pulmonary comorbidities, COPD, renal comorbidities, acute renal failure, neurological comorbidities, obesity, cachexia, ECOG > 2, Davies score > 2, Charlson score > 2, age-adjusted Charlson score > 9, sepsis, anemia, abnormal values of WBC, platelets, glycemia, creatinine, electrolyte disturbances, acidosis, coagulation disorders, ASA > 2, operative time > 120 min, and peritonitis. Many of these risk factors are found as predictors of other risk scores for OCC patients with emergency surgery. We did not confirm the type of operation or synchronous metastases as risk factors for postoperative mortality.

The five factors of the AFC/OCC score were also confirmed as risk factors for postoperative mortality by univariate logistic regression analysis, with small differences in the OR values: for age ≥ 75 (OR = 3.34 versus 2.35 in the AFC/OCC score training group OCC), hemodynamic failure (OR = 6.79 versus 10.62), pulmonary comorbidities (OR = 5.76 versus 3.45), ASA score ≥ 3 (OR = 6.97 versus 2.14), and proximal tumors (OR = 2.952, compared to 1.86) [21].

The nine independent predictors identified within a multivariate logistic analysis model were: age > 70 years, CHF, ECOG > 2, sepsis, obesity or cachexia, abnormal values of platelets or creatinine, and proximal tumors.

Advanced age is a well-known risk factor for postoperative death in colorectal emergency surgery [15,17,41,42,45]. In our study, the postoperative mortality predictor of age > 70 years presented OR = 12.9, one of the strongest predictors of the model.

Among all comorbidities, CHF seems to be implicated as a cause of death after surgery in these patients; one out of three patients with CHF died postoperatively (6/18). This predictor (OR = 13.5) proves to be a strong predictive factor in the model, which was also identified in another study on CRC patients [17].

The functional status assessed by the ECOG grade > 2 (ECOG 3—capable of only limited self-care; ECOG 4—completely disabled) was a predictor of postoperative mortality also identified by a model that included nonagenarians [46] and in another score for emergency surgery [47].

Sepsis (OR = 14.0) proved to be the strongest predictor of postoperative mortality. There were 18 deaths out of 40 patients with sepsis present at hospital admission, and this was one of the predictors of the SRC score but also of other studies [10,46].

We identified obesity as a predictor in the model: one out of four obese patients died in the first 30 postoperative days (9/36). Contrary to our results, a recent study did not identify significant differences in postoperative mortality between obese and normal-weight patients with OCC [48].

Another predictor identified was cachexia; approximately one out of three cachectic patients died (30/96). Other studies also identified malnutrition as a mortality risk factor [17,47].

The abnormal value of platelets, with OR = 11.6, is the fourth predictor of the model, being identified by other authors as a risk factor [40,47]. The abnormal value of serum creatinine was another predictor identified in the model, which is also similar to other authors [36,46,47,49].

We found, like Manceau in his study, that proximal tumors within the colon were predictors of postoperative mortality; however, other studies that did not identify this association [50,51].

The model has an excellent discrimination power (AUC = 0.947)—it identifies the high-risk group of patients with an estimated mortality rate > 14%—and a good calibration (H-L test > 0.05). The performance of the own model is better than the performance of other models [17,21,52]. Although we initially developed this model by analyzing all potential preoperative or intraoperative predictors likely to influence the postoperative course, we found that only some preoperative predictors were included in the model. This finding is relevant because it shows that the patient’s age and biological status, including septic status before surgery, are predictors of postoperative mortality rather than factors related to the surgical procedure. The risk score developed based on this model shows an excellent performance (AUC = 0.932) and a good calibration comparable to other existing scores for predicting postoperative mortality [52].

It provides a stratification of the risk of postoperative mortality in four groups depending on the number of factors present in each patient, similarly to other scores [15,17,20]. The nine factors of the score are routinely evaluated in any patient admitted to the hospital. If age, functional status, cachexia, or obesity and the proximal location of the tumor are non-correctable, the other four can be corrected if the surgical intervention can be delayed. Although the AFC/OCC score is easier to remember, as it has only five factors compared to the nine of the proposed score, the new score contains more possibly correctable factors (4) compared to the two of the French score—pulmonary comorbidities and hemodynamic instability.

Our score identifies especially those who have a high risk of dying, with the prediction of 100% mortality being reached in patients presenting at least 6/9 factors, unlike the French score, which has a particularly good prediction of survivors, while patients with all five present factors have a prediction of postoperative mortality of only 50% [21].

Therefore, it could be known and applied by the surgeon before the discussion with the patient regarding the informed consent for the surgical intervention. If this score indicates an increased risk of postoperative mortality, the patient’s management should be directed towards a non-surgical procedure, for example, the use of an endoscopic colonic stent or other non-surgical procedures such as a decompression tube insertion in patients with malignant left colonic obstruction. Overcoming the acute phase can allow the patient to undergo a programmed intervention after a possible correction of some risk factors, such as abnormal values of platelets and creatinine, CHF, or sepsis criteria.

The limits of the study are related to the retrospective nature, the size of the group, as well as the number of events and the single center of research. Selection bias and the limitations of the observational design study are part of the limitations of the study. To obtain a high-performance model, we used the entire group of patients as a training group, preferring to do the internal validation through the bootstrapping method. Limitations related to the long period of inclusion of cases are driven by possible improvements in health care, which could lead to different outcomes in later years. However, unfortunately, postoperative mortality rates were not significantly different over the whole period. In addition, we believe that this timing issue has a limited effect on our findings because except for sepsis, all predictors of postoperative mortality were unchanged over the study years. During the study period, the definition of sepsis underwent some changes; the classification in the three categories, i.e., SIRS (systemic inflammatory response syndrome), sepsis, and septic shock, required the interpretation of the existing data to assess this predictor.

A statistical limitation is the use of the H-L test, which has various flaws, but some authors find it acceptable if the study group > 400 patients. Another limitation would be related to the fact that we assigned equal weights to each risk factor in the own model without taking into account the risk coefficients identified by the model. The decision was made so that it would become as easy as possible to remember and apply the score by the surgeon, which was confirmed after we found that it does not greatly alter the statistical performance (AUC = 0.947 model, AUC = 0.932 score).

This score could provide reliable assistance in surgical decision making and optimization of peri-operative care. Although the score is internally validated, it is necessary to verify its performance and validate this score on prospective batches or external validation sample.

## 5. Conclusions

This retrospective study was conducted on an external population of patients with emergency surgery for OCC to validate the SRC, proposed by the ACS, and the score of postoperative mortality after emergency surgery for OCC, proposed by the AFC. SRC demonstrated a strong power of discrimination in the prediction of postoperative mortality and good calibration. Along with the confirmation of the performance of the prediction, another advantage of its application is obtaining a personalized risk, as these are elements that explain its reliability and effectiveness in current practice.

For the AFC/OCC score, we obtained an adequate power of discrimination in the prediction of mortality and a reasonable calibration; the easy memorization of the score factors and their simple applications are arguments that support its application to patients with OCC.

The stratification of the risk of postoperative mortality with the help of the Mortality Score—Clinic County Emergency Hospital of Galati/OCC could provide reliable assistance in making surgical management decisions (immediate surgical intervention or its postponement to reduce this risk by correcting the risk factors of the score), optimizing postoperative care and adequate information of patients and family members. The prediction of 100% mortality in a patient with at least six of the nine factors of the score recommends this score as a good positive predictor of postoperative mortality, which can be especially used for the selection of patients for immediate palliative surgical interventions (DC, BP) or in two-step surgical procedures (HP) or the use of non-surgical procedures in those with very high risk.

The easy identification of the score factors, the possibility of preoperative correction of some of them (abnormal values of platelets or creatinine, sepsis criteria, or improvement of CHF), and the statistical performance obtained in the training group recommend the Mortality Score—Clinic County Emergency Hospital of Galați/OCC for clinical use, but of course, this requires validation on prospective groups or external validation groups.

## Figures and Tables

**Figure 1 ijerph-19-13513-f001:**
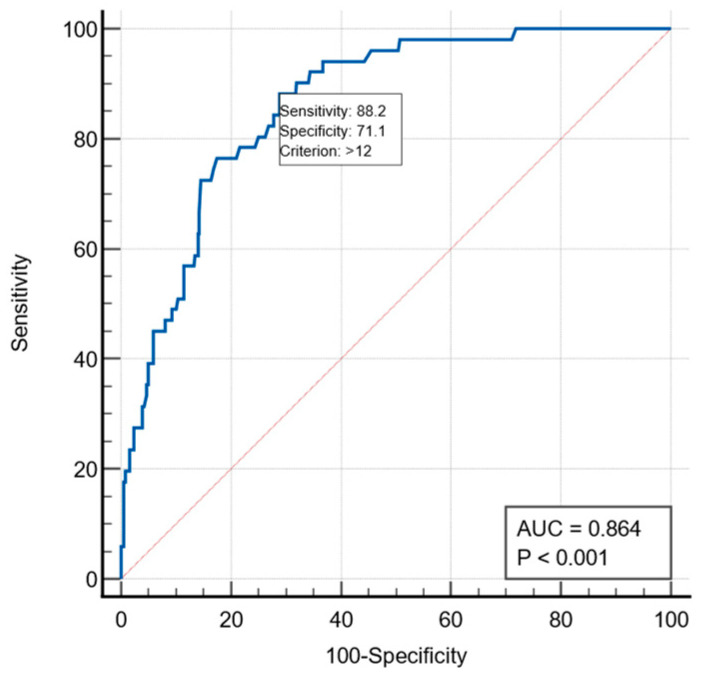
ROC curve—discrimination accuracy of the mortality prediction of the SRC.

**Figure 2 ijerph-19-13513-f002:**
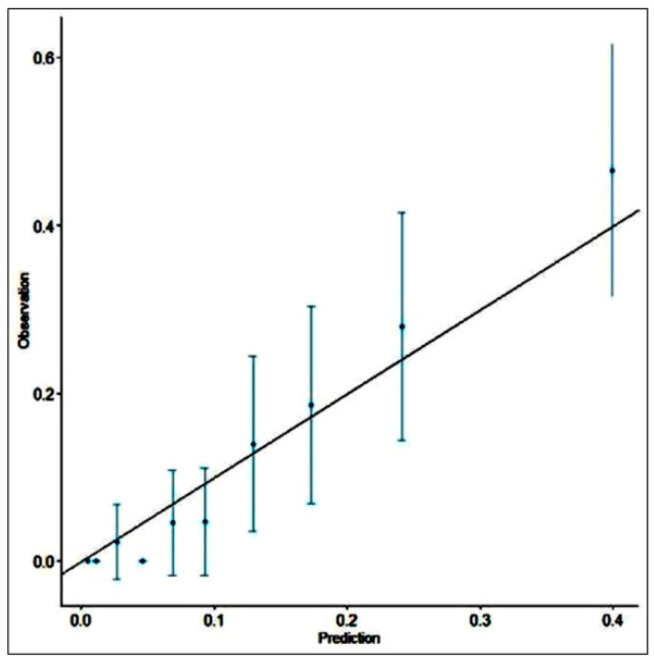
Calibration graph of the mortality prediction of the SRC.

**Figure 3 ijerph-19-13513-f003:**
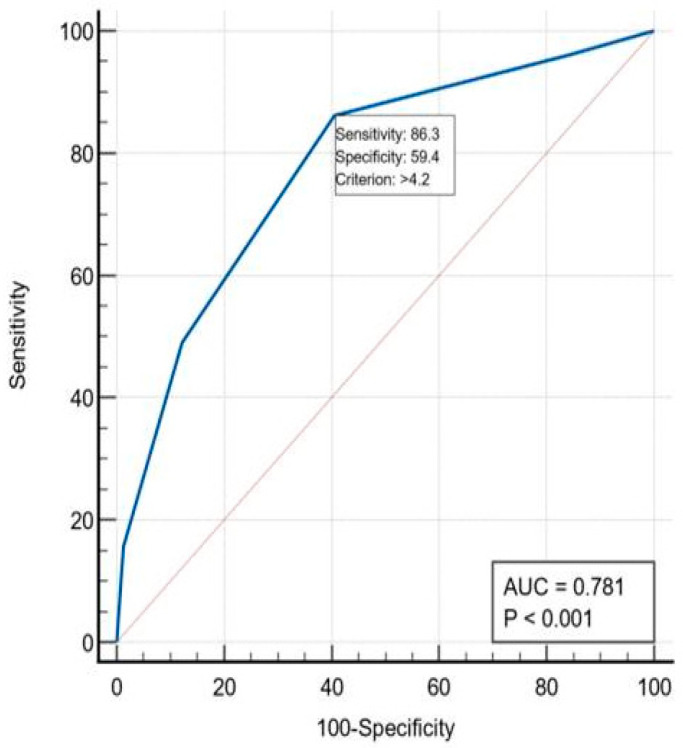
ROC curve—discrimination accuracy of the mortality prediction of the AFC/OCC score.

**Figure 4 ijerph-19-13513-f004:**
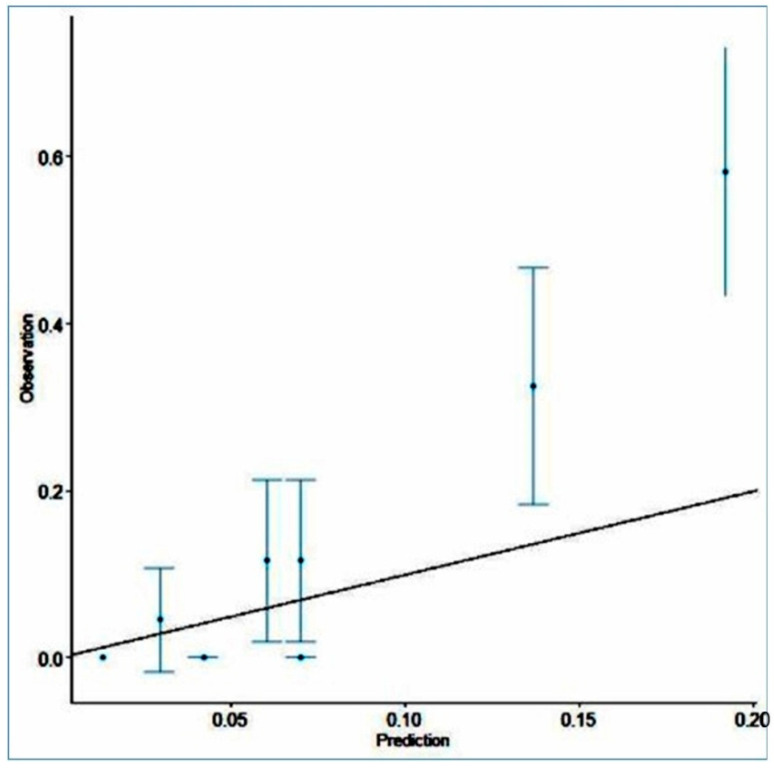
Calibration graph of the mortality prediction of the AFC/OCC score.

**Figure 5 ijerph-19-13513-f005:**
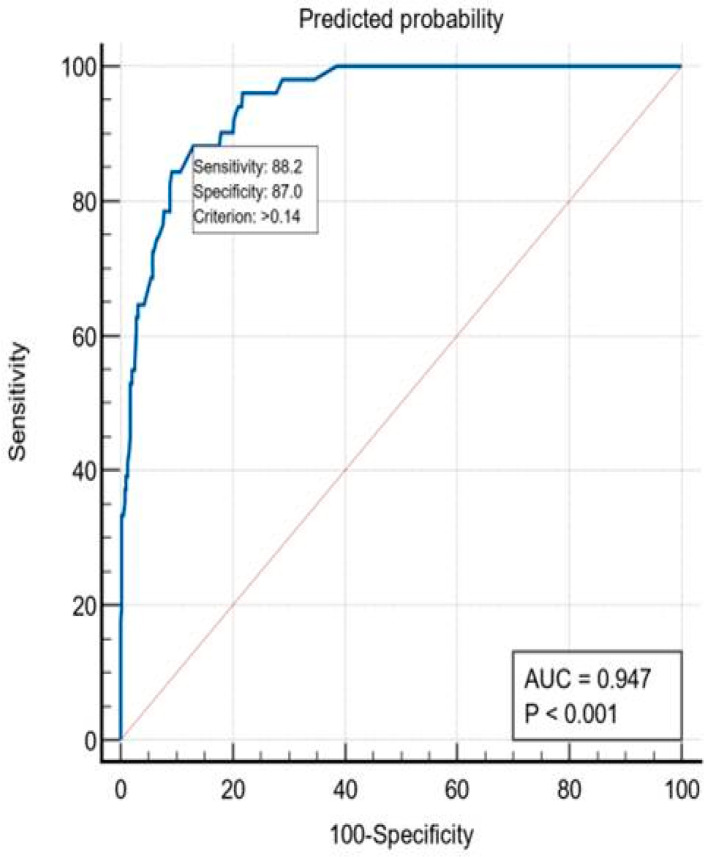
ROC curve of the multivariate logistic regression model.

**Figure 6 ijerph-19-13513-f006:**
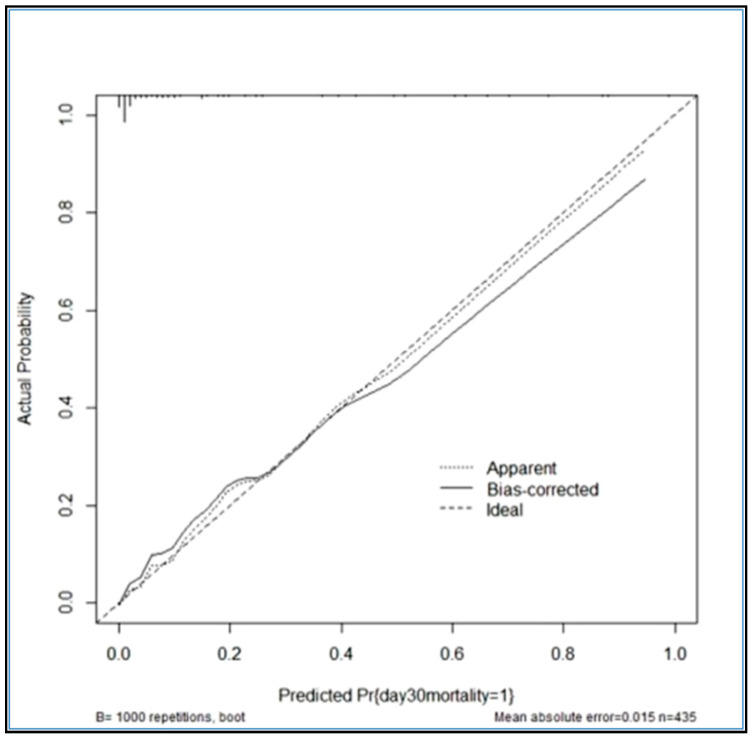
Model calibration graph by bootstrapping method.

**Figure 7 ijerph-19-13513-f007:**
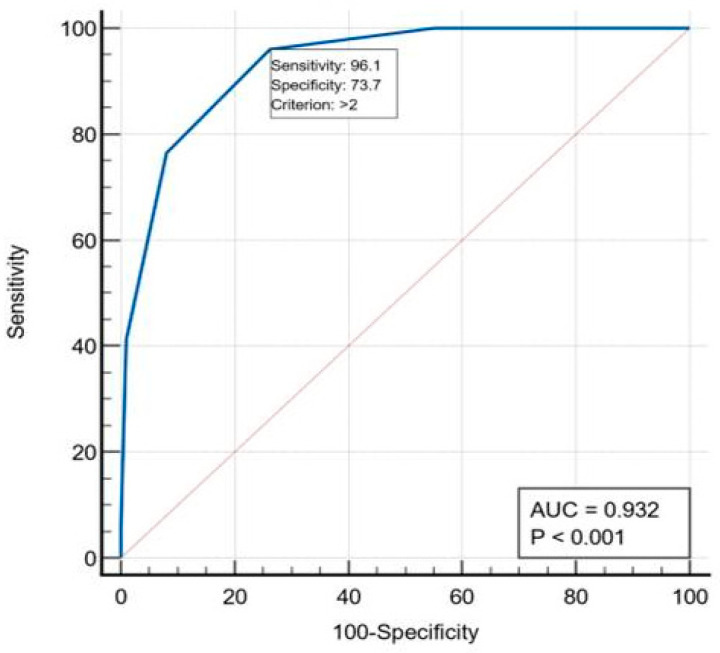
ROC curve—Mortality Score—Clinic County Emergency Hospital of Galați/OCC.

**Table 1 ijerph-19-13513-t001:** Descriptive data and statistic association with postoperative mortality.

	30 Days Mortality = NO	30 Days Mortality = YES	*p-*Value (Test)
Age > 70	178/384 (46.4%)	41/51 (80.4%)	0.000005 (^1^)
Age ≥ 75	115/384 (29.9%)	30/51 (58.8%)	0.000040 (^1^)
Gender			0.389579 (^1^)
F	152/384 (39.6%)	17/51 (33.3%)	
M	232/384 (60.4%)	34/51 (66.7%)	
History of surgery—yes	123/384 (32%)	23/51 (45.1%)	0.063356 (^1^)
Cardiovascular comorbidities—yes	138/384 (35.9%)	21/50 (42%)	0.402639 (^1^)
HTN—yes	100/384 (26%)	11/51 (21.6%)	0.491178 (^1^)
CHF = yes	12/384 (3.1%)	6/51 (11.8%)	0.011893 (^2^)
Hemodynamic failure—yes	51/384 (13.3%)	26/51 (51%)	0.000000 (^1^)
Pulmonary comorbidities—yes	12/384 (3.1%)	8/51 (15.7%)	0.000869 (^2^)
COPD—yes	2/384 (0.5%)	2/51 (3.9%)	0.069324 (^2^)
Dyspnea-yes	5/384 (1.3%)	2/51 (3.9%)	0.192998 (^2^)
Renal comorbidities—yes	25/384 (6.5%)	11/51 (21.6%)	0.001230 (^2^)
AKF—yes	1/384 (0.3%)	2/51 (3.9%)	0.037464 (^2^)
Neurological comorbidities—yes	16/384 (4.2%)	9/51 (17.6%)	0.000982 (^2^)
Hepatic comorbidities—yes	15/384 (3.9%)	5/51 (9.8%)	0.071639 (^2^)
Diabetes mellitus—yes	39/384 (10.2%)	7/51 (13.7%)	0.436107 (^1^)
Smoker—yes	57/384 (14.8%)	9/51 (17.6%)	0.600069 (^1^)
Obesity—yes	27/384 (7%)	9/51 (17.6%)	0.025106 (^2^)
Cachexia—yes	66/384 (17.2%)	30/51 (58.8%)	0.000000 (^1^)
ECOG > 2	80/384 (20.8%)	30/51 (58.8%)	0.000000 (^1^)
Davies score > 2	63/384 (16.4%)	20/51 (39.2%)	0.000098 (^1^)
Charlson score > 2	238/384 (62%)	47/51 (92.2%)	0.000020 (^1^)
Age-adjusted Charlson score > 9	197/384 (51.3%)	45/51 (88.2%)	0.000000 (^1^)
Sepsis—yes	22/384 (5.7%)	18/51 (35.3%)	0.000000 (^2^)
WBC—aN	153/384 (39.8%)	44/51 (86.3%)	0.000000 (^1^)
Anemia—yes	236/384 (61.5%)	44/51 (86.3%)	0.000507 (^1^)
Platelets—aN	37/384 (9.6%)	26/51 (51%)	0.000000 (^1^)
Glycemia—aN	76/384 (19.8%)	22/51 (43.1%)	0.000177 (^1^)
Creatinine—aN	138/384 (35.9%)	37/51 (72.5%)	0.000000 (^1^)
Electrolyte disturbances—yes	102/384 (26.6%)	31/51 (60.8%)	0.000000 (^1^)
Acidosis—yes	69/384 (18%)	27/51 (52.9%)	0.000000 (^1^)
Coagulation disorders—yes	36/384 (9.4%)	18/51 (35.3%)	0.000000 (^1^)
Pneumoperitoneum—yes	17/384 (4.4%)	5/51 (9.8%)	0.161210 (^1^)
ASA > 2	241/384 (62.8%)	47/51 (92.2%)	0.000030 (^1^)
Operative time > 120 min	91/384 (23.7%)	25/51 (49%)	0.000122 (^1^)
Operative time			0.000313 (^1^)
<150 min	294/384 (76.6%)	27/51 (52.9%)	
≥150 min	90/384 (23.4%)	24/51 (47.1%)	
Peritonitis—yes	13/384 (3.4%)	5/51 (9.8%)	0.047547 (^2^)
Tumor site			0.518164 (^3^)
C	37/384 (9.6%)	7/51 (13.7%)	
A	11/384 (2.9%)	3/51 (5.9%)	
HF	29/384 (7.6%)	3/51 (5.9%)	
T	37/384 (9.6%)	4/51 (7.8%)	
SF	37/384 (9.6%)	5/51 (9.8%)	
D	41/384 (10.7%)	5/51 (9.8%)	
S	133/384 (34.6%)	21/51 (41.2%)	
RSJ	59/384 (15.4%)	3/51 (5.9%)	
Site of the tumor			0.540466 (^1^)
distal	272/384 (70.8%)	34/51 (66.7%)	
proximal	112/384 (29.2%)	17/51 (33.3%)	
Ischemic lesions of the distended colon—yes	20/384 (5.2%)	3/51 (5.9%)	0.742619 (^2^)
Synchronous metastasis—yes	78/384 (20.3%)	15/51 (29.4%)	0.136440 (^1^)
Macroscopic invasion of the adjacent organs—yes	73/384 (19%)	10/51 (19.6%)	0.918744 (^1^)
Stoma performed—yes	218/384 (56.8%)	32/51 (62.7%)	0.453890 (^2^)
Surgery			0.639195 (^3^)
PCA	127/384 (33.1%)	13/51 (25.5%)	
PCAI	5/384 (1.3%)	2/51 (3.9%)	
TCA	36/384 (9.4%)	6/51 (11.8%)	
HP	65/384 (16.9%)	9/51 (17.6%)	
BP	3/384 (0.8%)	0/51 (0%)	
DC	148/384 (38.5%)	21/51 (41.2%)	
Intraoperative complications—yes	5/384 (1.3%)	2/51 (3.9%)	0.192998 (^2^)
ACS-NSQIP score	6.60 (1.90, 14.45)	24.40 (17.5,38.4)	0.000000 (^4^)
AFC/OCC score	4.20 (4.20, 7.00)	7.00 (7.00,16.90)	0.000000 (^4^)

(^1^), Pearson chi-square; (^2^), Fisher’s exact test; (^3^) likelihood ratio; (^4^), Mann–Whitney test; HTN, arterial hypertension; CHF, congestive heart failure; COPD, chronic obstructive pulmonary disease; AKF, acute kidney failure; ASA, American Society of Anesthesiologists; ECOG, Eastern Cooperative Oncology Group performance status; WBC, white blood cell; C, cecum; A, ascending colon; HF, hepatic flexure of colon; T, transverse colon; SF, splenic flexure of colon; D, descending colon; S, sigmoid; RSJ, rectosigmoid junction; PCA, partial colectomy with anastomosis; PCAI, partial colectomy with anastomosis with protective ileostomy; TCA, total colectomy with ileoproctostomy; HP, colic resection with stoma; BP, internal by-pass; DC, diverting colostomy; aN, abnormal value.

**Table 2 ijerph-19-13513-t002:** Univariate and multivariate logistic regression analysis.

	Univariate Logistic Regression		Multivariate Logistic Regression	
	*p-*Value	OR (95%CI)	*p-*Value	OR (95%CI)
Age > 70	0.000022	4.75 (2.31, 9.76)	0.000079	12.9 (3.62, 46)
Age ≥ 75	0.000079	3.34 (1.83, 6.08)		
Gender F M	0.390617 (overall)		-	
History of surgery—yes	0.065721		-	
Cardiovascular comorbidities—yes	0.403536		-	
HTN—yes	0.492044		-	
CHF—yes	0.006799	4.13 (1.47, 11.6)	0.000183	13.5 (3.45, 52.5)
Hemodynamic failure—yes	0.000000	6.79 (3.64, 12.7)	-	
Pulmonary comorbidities—yes	0.000000	5.76 (2.23, 14.9)	-	
COPD—yes	0.042320	7.79 (1.07, 56.6)	-	
Dyspnea—yes	0.184101		-	
Renal comorbidities—yes	0.000565	3.94 (1.80, 8.62)	-	
AKF—yes	0.025893	15.6 (1.39, 176)	-	
Neurological comorbidities—yes	0.000363	4.92 (2.05, 11.8)	-	
Smoker—yes	0.600601		-	
Obesity—yes	0.012729	2.83 (1.24, 6.42)	0.003193	6.80 (1.90, 24.3)
Cachexia—yes	0.000000	6.88 (3.71, 12.8)	0.000399	5.19 (2.08, 12.9)
ECOG > 2	0.000000	5.42 (2.95, 9.98)	0.000056	6.69 (2.65, 16.9)
Davies score > 2	0.000184	3.28 (1.76, 6.13)	-	
Charlson > 2	0.000201	7.20 (2.54, 20.4)	-	
Age-adjusted Charlson score > 9	0.000011	7.12 (2.96, 17.1)	-	
Sepsis—yes	0.000000	8.97 (4.37, 18.4)	0.000017	14.0 (4.21, 46.9)
WBC—aN	0.000000	9.49 (4.16, 21.6)	-	
Anemia—yes	0.001098	3.94 (1.73, 9)	-	
Platelets—aN	0.000000	9.75 (5.11, 18.6)	0.000001	11.6 (4.30, 31.3)
Glycemia—aN	0.000296	3.07 (1.67, 5.64)	-	
Creatinine—aN	0.000003	4.71 (2.46, 9.01)	0.006092	3.39 (1.41, 8.13)
Electrolyte disturbances—yes	0.000003	4.28 (2.33, 7.85)	-	
Acidosis—yes	0.000000	5.13 (2.79, 9.43)	-	
Coagulation disorders—yes	0.000001	5.27 (2.70, 10.3)	-	
Pneumoperitoneum—yes	0.109037		-	
ASA > 2	0.000258	6.97 (2.46, 19.8)	-	
Operative time, min ≥ 120	0.000209	3.09 (1.70, 5.62)	-	
Peritonitis—yes	0.039192	3.10 (1.05, 9.09)	-	
Proximal tumor—yes	0.540927		0.021415	2.95 (1.17, 7.42)
Synchronous metastasis—yes	0.139388		-	
Stoma performed—yes	0.418318		-	
PCA PCAI BP DC TCA HP	0.721188 (overall)		-	
ACS-NSQIP score	0.000000	1.10 (1.07, 1.13)		
AFC/OCC score	0.000000	1.17 (1.11, 1.22)		

HTN, arterial hypertension; CHF, congestive heart failure; COPD, chronic obstructive pulmonary disease; AKF, acute kidney failure; ASA, American Society of Anesthesiologists; ECOG, Eastern Cooperative Oncology Group performance status; WBC, white blood cell; PCA, partial colectomy with anastomosis; PCAI, partial colectomy with anastomosis with protective ileostomy; TCA, total colectomy with ileoproctostomy; HP, colic resection with stoma; BP, internal by-pass; DC, diverting colostomy; aN, abnormal value.

**Table 3 ijerph-19-13513-t003:** Stratification of postoperative mortality—mortality rates according of risk factor numbers.

Number of Risk Factors	Deaths/Total Number	Mortality Rate (%)	OR	95%CI	*p-*Value
**0–2 (Ref)**	2/285	0.7%			0.000000
**3**	10/80	12.5%	20.2	4.33–94.3	0.000131
**4**	18/45	40.0%	94.3	20.8–428	0.000000
**5–6**	21/25	84.0%	743	128–4293	0.000000

Because of calculation procedure, the categories corresponding to values 0, 1, and 2 and 5 and 6 of the score have been pooled.

## Data Availability

Not applicable.

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
