# Peer review of "Mortality Risk Stratification in Emergency Surgery for Obstructive Colon Cancer—External Validation of International Scores, American College of Surgeons National Surgical Quality Improvement Program Surgical Risk Calculator (SRC), and the Dedicated Score of French Surgical Association (AFC/OCC Score)"

_ijerph, 2022, doi:10.3390/ijerph192013513_

Round 1
Reviewer 1 Report
Thank you for allowing me to review this paper which is of a great interest to Colon and Rectal Surgeons and to be able to discuss the predicted morbidity and mortality outcome to the patient and family before and after emergency surgery for large bowel obstruction due to tumor.
My wish is to have higher number of patients but the number used is statistically significant.
Author Response
Thank you for allowing me to review this paper which is of a great interest to Colon and Rectal Surgeons and to be able to discuss the predicted morbidity and mortality outcome to the patient and family before and after emergency surgery for large bowel obstruction due to tumor.
My wish is to have higher number of patients but the number used is statistically significant.
Submission Date
02 September 2022
Date of this review
19 Sep 2022 21:49:15
Response
Thank you for your kind permission to review this manuscript and to send your opinion in writing.
Reviewer 2 Report
Total resections with protective or permanent stomas are not reported.
The abbreviations for tumor sites require thorough clarification.
The criteria for obesity and sarcopenia should be thorougly reported.
Quite a long time span (10 years) for a retrospective study, taking under consideration , alterations been made in diagnostic criteria and therapeutic strategies and techniques during this time , possibly applied in these cases.
Author Response
- Reviewer’s comment -Total resections with protective or permanent stomas are not reported.
Response.
Thank you for your remark. These types of resections were not performed in the patients included in the study. We included 42 patients with total colectomy (TCA),(table 2), followed by ileoproctostomy without protective stoma in all patients.
- Reviewer’s comment - The abbreviations for tumor sites require thorough clarification.
Response
Thank you for your remark. I’ve added in the text more details in this regard and marked them in red.
We dichotomized the variable according to the anatomical segments of the colon into 8 groups: cecum(C), ascending colon (A), hepatic flexure of the colon (HF), transverse colon (T), splenic flexure of the colon (SF), descending colon (D), sigmoid (S), the rectosigmoid junction (RSJ). (p.4)
- Reviewer’s comment- The criteria for obesity and sarcopenia should be thorougly reported.
Response
Thank you for your remark. I’ve added in the text more details in this regard and marked them in red.
For the assessment of obesity and cachexia, we used the criteria recommended by the World Health Organization based on the Body Mass Index (BMI), obtained by dividing the body mass to the squared of the body height. (p.4)
Sarcopenia was not assessed in the study group.
- Reviewer’s comment - Quite a long time span (10 years) for a retrospective study, taking under consideration, alterations been made in diagnostic criteria and therapeutic strategies and techniques during this time, possibly applied in these cases.
Response
Thank you for your remark. I agree with your opinion, retrospective manner and long study period are part of the limitations of the study. I’ve added in the text more details in this regard and marked them in red.
Limitations related to the long period of inclusion of cases are driven by possible improvements in health care, which could lead to different outcomes in later years. But unfortunately, postoperative mortality rates were not significantly different over the whole period. In addition, we believe that this timing issue has a limited effect on our findings because except for sepsis, all predictors of postoperative mortality were unchanged over the study years.Another limitation would be the definition of sepsis which underwent some changes during the study period, the classification in the 3 categories: SIRS (systemic inflammatory response syndrome), sepsis and septic shock required the interpretation of the existing data. During the study period, the definition of sepsis underwent some changes; the classification in the 3 categories: SIRS (systemic inflammatory response syndrome), sepsis and septic shock required the interpretation of the existing data to asses this predictor. (p.18)
